# GraphSeq2Seq: Graph-Sequence-to-Sequence for Neural Machine Translation

## Abstract

Sequence-to-Sequence (Seq2Seq) neural models have become popular for text generation problems, e.g. neural machine translation (NMT) (Bahdanau et al., 2014; Britz et al., 2017), text summarization (Nallapati et al., 2017; Wang & Ling, 2016), and image captioning (Venugopalan et al., 2015; Liu et al., 2017). Though sequential modeling has been shown to be effective, the dependency graph among words contains additional semantic information, and thus can be utilized for sentence modeling. In this paper, we propose a Graph-Sequence-to-Sequence (GraphSeq2Seq) model to fuse the dependency graph among words into the traditional Seq2Seq framework. For each sample, the sub-graph of each word is encoded to a graph representation, which is then utilized to sequential encoding. At last, a sequence decoder is leveraged for output generation. Since above model fuses different features by contacting them together to encode, we also propose a variant of our model that regards the graph representations as additional annotations in attention mechanism (Bahdanau et al., 2014) by separately encoding different features. Experiments on several translation benchmarks show that our models can outperform existing state-of-the-art methods, demonstrating the effectiveness of the combination of Graph2Seq and Seq2Seq.

## 1 Introduction

Neural machine translation (NMT) is a hot topic in Natural Language Processing. Most of NMT models belong to the encoder-decoder framework (Sutskever et al., 2014; Cho et al., 2014) which encodes the source language to a representation vector and then decodes the vector to the target language. When both the input and output are sequential words, the models based on this framework are also called Sequence-to-Sequence (Seq2Seq) models (Sutskever et al., 2014; Bahdanau et al., 2014). More importantly, the dependency graph among words is also critical for sentence modeling in the machine translation, since it contains additional semantic information (Bastings et al., 2017). Even the sequential sentence may serialize closely-related elements far away, the dependency graph can involve all of the elements closely, which is helpful for sentence modeling. However, most of existing Seq2Seq models do not well exploit this dependency graph information. How to effectively combine the dependency graph and the sequential information to improve the translation quality, therefore, becomes a challenging problem.

To address this problem, we propose a Graph-Sequence-to-Sequence (GraphSeq2Seq) model by fusing the dependency graph among words into the Seq2Seq framework. For each sentence, we first utilize a dependency parser to obtain the dependency graph among words. Each sentence has a graph and each word has a sub-graph. Then the sub-graph is embedded to a representation for the word, and it is passed to a bidirectional sequence encoder. Finally, a decoder with attention mechanism generates the outputs. Since above model fuses different features by contacting them together to encode, we propose a variant of our model that tries another strategy by separately encoding different features. We evaluate GraphSeq2Seq model and its variant model on several machine translation benchmarks. Experiment results show our models outperforms existing state-of-the-art methods, demonstrating the effectiveness of the combination of Graph2Seq and Seq2Seq.

The main contributions of this paper are as follows.

- We propose a GraphSeq2Seq model by combining Graph2Seq model and Seq2Seq model. The dependency graph information and sequential information are helpful for sentence modeling to generate high-quality translations.

- Our GraphSeq2Seq model passes the embedding of the sub-graph of a word to a bidirectional sequence encoder. Besides, a variant of GraphSeq2Seq model is also deployed which leverages features separately to calculate the context vectors in attention mechanism.

- Experiment results on four translation benchmarks show the better performance of our models than state-of-the-arts, demonstrating the effectiveness of combining dependency graph information and sequence information.

## 2 BACKGROUND

We consider the problem that this paper is trying to address as how to combine Seq2Seq model and Graph2Seq model to improve the translation quality. Given an input sentence $X = \{x_1, x_2, ..., x_n\}$ and its dependency graph for each word $G = \{g_1, g_2, ..., g_n\}$, our models combine $X$ and $G$ to generate the output sequence $Y = \{y_1, y_2, ..., y_m\}$. Three sets of input-output pairs $(X, Y)$ are assumed to be available for training, validating and testing. The graph $G$ is parsed from the sentence $X$. The trained model is evaluated by computing the average task-specific score $R(\hat{Y}, Y)$ on the test set, where $\hat{Y}$ is the prediction.

### 2.1 SEQUENCE-TO-SEQUENCE MODELS

Seq2Seq Models (Bahdanau et al., 2014; Neubig, 2017) are general based on Recurrent Neural Networks (RNN) encoder-decoder framework (Sutskever et al., 2014; Cho et al., 2014). Seq2Seq models is widely used for the Machine Translation task, but it has been also used for a variety of other tasks, including Summarization, Conversational Modeling, and Image Captioning etc. Rush et al. (2015) utilized sequence-to-sequence encoder-decoder Long Short-Term Memory (LSTM) with attention to train a neural model for summarization task. Shang et al. (2015) proposed a neural network-based response generator for Short Text Conversation using the encoder-decoder framework. As long as the problem can be phrased as encoding input data in one format and decoding it into another format, Seq2Seq can be utilized to address it.

Given a sequence $X$, an encoder reads it into hidden state vector $h$. Generally, a bidirectional RNN (Schuster & Paliwal, 1997) is utilized. For the input sequence with ordering from $x_1$ to $x_T$, the forward RNN calculates a sequence of its forward hidden states $\{\overrightarrow{h_1}, \cdots, \overrightarrow{h_T}\}$. Meanwhile, reversing the input as the order from $x_T$ to $x_1$, the backward RNN calculates a sequence of its backward hidden states $\{\overleftarrow{h_1}, \cdots, \overleftarrow{h_T}\}$. Then we obtain the final hidden states by concatenating them as $h_j = \{\overrightarrow{h_j}, \overleftarrow{h_j}\}$ which saves the summaries of both the preceding words and the following words.

Attention mechanism (Bahdanau et al., 2014) aims to find the parts of inputs that should be focused. Thus, the context vector $c$ is calculated by a weighted sum of the final hidden states:

$$c_t = \sum_{j=1}^{T} \alpha_{t,j} h_j, \tag{1}$$

where the weight $\alpha_{t,j}$ is computed by an alignment model. Its details can be found in Bahdanau et al. (2014).

Given the predicted preceding words $\{y_1, y_2, \cdots, y_{t-1}\}$, context vector $c_t$, and the RNN hidden state $s_t$, the decoder calculates a probability of the next word $y_t$:

$$p(y_t | X, y_1, y_2, \cdots, y_{t-1}) = g(y_{t-1}, s_t, c_t), \tag{2}$$

where $g(\cdot)$ is a softmax activation function and

$$s_t = f(y_{t-1}, s_{t-1}, c_t), \tag{3}$$

here, $f$ is a non-linear function. It can be a logistic function, an LSTM unit, or a Gated Recurrent Unit (GRU).

Recently, Byte Pair Encoding (BPE) is utilized for word segmentation Sennrich et al. (2016) to overcome the out-of-vocabulary problem and rare in-vocabulary problem. Both of Edunov et al. (2018) and Deng et al. (2018) achieve state-of-the-art results, where they utilize BPE on both source side and target side. However, our GraphSeq2Seq cannot apply BPE on the source side. Thus, we do not choose this kind of methods for comparison.

## 2.2 GRAPH-TO-SEQUENCE MODELS

Graph2Seq models (Cohn et al., 2018; Gildea et al., 2018) are generally utilized to address the generation of graph-structured data (such as Abstract Meaning Representation (AMR) (Banarescu et al., 2013) and dependency graph (Chen et al., 2017)) to text. It is to recover a text representing the same meaning as an input graph. Flanigan et al. (2016) converted input graphs to trees by splitting re-entrances, and then translated the trees into sentences with a tree-to-string transducer. Graph convolutional networks (GCN) (Kipf & Welling, 2016) are used for semantic role labeling (Marcheggiani & Titov, 2017) and neural machine translation (Bastings et al., 2017). Graph state LSTM (Gildea et al., 2018; Song et al., 2018) adopts gated operations for making updates, while GCN uses a linear transformation.

A graph consists of triples $(i, j, l)$, where $i$ and $j$ are indices of the incoming and outgoing nodes, $l$ is their edge label. For a sub-graph $g(n)$ of the $n$-th node $x_n$, Gildea et al. (2018) described incoming representation and outgoing representation as:

$$x_n^{in} = \sum_{(i,\ n,\ l) \in g^{in}(n)} W[x_i,\ e_l] + b, \tag{4}$$

$$x_n^{out} = \sum_{(n,\ j,\ l) \in g^{out}(n)} W[x_j,\ e_l] + b, \tag{5}$$

where $x_i$ is the node representation and $e_l$ is the edge representation, $[\cdot]$ is a concat operation for them. $W$ and $b$ are the weight and bias to encode the representations. Then Gildea et al. (2018) adopted a graph state LSTM to encode each sub-graph as:

$$i_n(t) = \sigma(W_i x_n^{in} + \hat{W}_i x_n^{out} + U_i h_n^{in} + \hat{U}_i h_n^{out} + b_i), \tag{6}$$

$$o_n(t) = \sigma(W_o x_n^{in} + \hat{W}_o x_n^{out} + U_o h_n^{in} + \hat{U}_o h_n^{out} + b_o), \tag{7}$$

$$f_n(t) = \sigma(W_f x_n^{in} + \hat{W}_f x_n^{out} + U_f h_n^{in} + \hat{U}_f h_n^{out} + b_f), \tag{8}$$

$$u_n(t) = \sigma(W_u x_n^{in} + \hat{W}_u x_n^{out} + U_u h_n^{in} + \hat{U}_u h_n^{out} + b_u), \tag{9}$$

$$c_n(t) = f_n(t) \odot c_n(t-1) + i_n(t) \odot u_n(t), \tag{10}$$

$$h_n(t) = o_n(t) \odot tanh(c_n(t)), \tag{11}$$

where $i_n(t)$, $o_n(t)$ and $f_n(t)$ are the input, output and forget gates. $W$, $\hat{W}$, $U$, $\hat{U}$ and $b$ are model parameters. Besides, the incoming hidden and outgoing hidden also consider the graph structure so that they are represented by $\sum_{(i,\ n,\ l) \in g^{in}(n)} h_i(t-1)$ and $\sum_{(n,\ j,\ l) \in g^{out}(n)} h_j(t-1)$. Then the hidden vectors are adopted in the decoder.

## 3 GRAPH-SEQUENCE-TO-SEQUENCE MODEL

This section introduces the details about our GraphSeq2Seq model. Fig. 1 is an overview of our model. Given an input sentence $X$, GraphSeq2Seq 1) gets the sub-graph $g_n$ including the incoming nodes $x_i$ and outgoing nodes $x_o$ for each word $x_n$, and uses a graph state LSTM (Gildea et al., 2018) to encode $g_n$; 2) fuses the word representation, sub-graph state, incoming and outgoing representations into a full graph representation; 3) regards the graph representation as the input of a bidirectional sequence encoder (Schuster & Paliwal, 1997); 4) with an attention mechanism (Bahdanau et al., 2014), the decoder generates the output words $Y$.

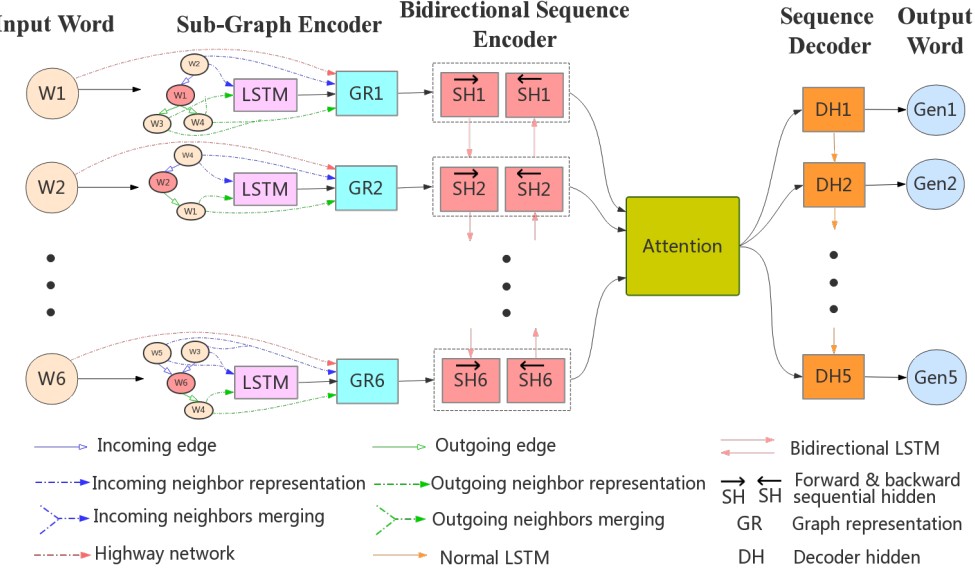

Figure 1: An overview for our GraphSeq2Seq model by combining Graph2Seq (Gildea et al., 2018) and Seq2Seq (Bahdanau et al., 2014).

### 3.1 SUB-GRAPH ENCODER

Given an input sentence $X$, we use a dependency parser Spacy[1] which is a free open-source library for Natural Language Processing in Python to extract the dependency graph $G$ among words. For each word, there is exactly another one word corresponding to it. Then we get a triple $(i, j, l)$, where $i$ and $j$ are indices of the source (incoming) and target (outgoing) nodes, $l$ is their dependency which is seen as the edge label in the graph. So each word has a sub-graph consisting of several triples from the whole graph. In the sub-graph $g(n)$ of the $n$-th node $x_n$, incoming representation and outgoing representation are $x_n^{in}$ and $x_n^{out}$ calculated by Eq. 4 and Eq. 5. Then through the sub-graph encoder, we get

$$h_n(t), c_n(t) = gsLSTM(x_n^{in}, x_n^{out}, h_n^{in}, h_n^{out}, c_n(t-1)), \tag{12}$$

where gsLSTM is the graph state LSTM (Gildea et al., 2018) described in Section 2.2, and the incoming hidden $h_n^{in}$ and outgoing hidden $h_n^{out}$ also consider the graph structure with different weights so that

$$h_n^{in}(t) = \sum_{(i,\ n,\ l) \in g^{in}(n)} w_{i,n} * h_i(t-1), \tag{13}$$

$$h_n^{out}(t) = \sum_{(n,\ j,\ l) \in g^{out}(n)} w_{n,j} * h_j(t-1). \tag{14}$$

The final incoming hidden $h_n^{in}$, outgoing hidden $h_n^{out}$, and the final sub-graph hidden $h_n$ contains different information. We concat them trying to build a final graph representation. Since these hidden features may loss some information of the initial node representation, we adopt a highway network (Srivastava et al., 2015) to transform and keep the initial node representation $x_n$ by

$$H_n = (W_t x_n + b_t) * (W_c x_n + b_c) + x_n(1 - (W_c x_n + b_c)), \tag{15}$$

where $H_n$ is the output of the highway network which not only contains the transformed information but also carries the initial information. Finally, through the sub-graph encoder, we get the final representation which is a concat as

$$r_n = [h_n^{in},\ h_n^{out},\ h_n,\ H_n]. \tag{16}$$

### 3.2 BIDIRECTIONAL SEQUENCE ENCODER AND THE DECODER WITH ATTENTION

Aforementioned final representation $r$ for each node is utilized as a sequence input of a bidirectional LSTM (Schuster & Paliwal, 1997) encoder as traditional Seq2Seq models (Bahdanau et al.,

---

[1]https://spacy.io

2014). Given an representation sequence with ordering from $r_1$ to $r_n$, the forward hidden state and backward hidden state of $r_n$ is

$$\overrightarrow{h_n} = LSTM_f(r_n, \overrightarrow{h}_{n-1}), \tag{17}$$

$$\overleftarrow{h_n} = LSTM_b(r_n, \overleftarrow{h}_{n-1}), \tag{18}$$

where $LSTM_f$ is a forward LSTM and $LSTM_b$ is a backward LSTM. Note that, for the backward LSTM, we feed the reversed input as the order from $x_n$ to $x_1$. Then we obtain the final hidden states by concatenating them as $h_j = \{\overrightarrow{h_j}, \overleftarrow{h_j}\}$ which saves the summaries of both the preceding words and the following words. Finally, a decoder with the attention mechanism (Bahdanau et al., 2014) described in Section 2.1 is leveraged to generate outputs.

### 3.3    A VARIANT MODEL

Aforementioned GraphSeq2Seq model utilizes the output of the graph encoder to be the input of sequence encoder. However, the output of the graph encoder consists of different features, such as the hidden feature $h_n^{in}$ for incoming sub-graph, the hidden feature $h_n^{out}$ for outgoing sub-graph, and the node representations $x_n$. The original GraphSeq2Seq model directly concats these features to encode them together, which may lost some information. Here, we try to reserve the information by encoding these features separately. The main difference is that the original model is to encode the concated representation of the sub-graph by using only one Bi-LSTM, while the variant model leverages three Bi-LSTMs to respectively encode the specific representations of the sub-graph, including the incoming feature, the outgoing feature, and the node representation.

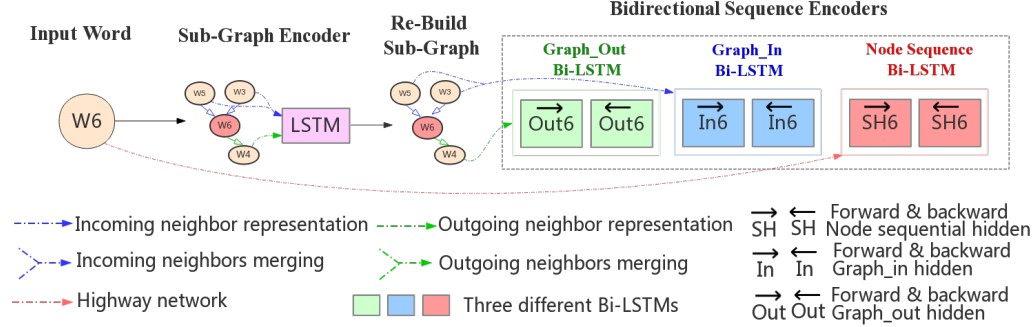

Figure 2: A variant for the encoder part of GraphSeq2Seq model. It encodes the features separately with three different Bi-LSTMs, while the original model is to encode the features by using only one Bi-LSTM.

As shown in Fig. 2, after the sub-graph encoder, we get the hidden feature for each node. Then we rebuild the sub-graph for the current node. For the rebuilded sub-graph, its outgoing hidden feature is the input of a Graph_Out Bi-LSTM, while its incoming hidden feature is for a Graph_in Bi-LSTM. For the current node, its node representation is utilized for a Node Sequence Bi-LSTM. Thus, final hidden states are the concat $[\overrightarrow{h_n^{out}}, \overleftarrow{h_n^{out}}, \overrightarrow{h_n^{in}}, \overleftarrow{h_n^{in}}, \overrightarrow{H_n}, \overleftarrow{H_n}]$. After that, a traditional attention mechanism (Bahdanau et al., 2014) as presented in Section 2.1 is utilized to generate outputs.

## 4    EXPERIMENTS

In this section, we evaluate our model on IWSLT 2014 German-to-English, IWSLT 2014 English-to-German (Cettolo et al., 2014), IWSLT 2015 English-to-Vietnamese (Cettolo et al., 2015), and WMT 2016 English-to-Czech (DBL, 2016) machine translation benchmarks. This section describes the details of the experiments, including the detailed implementations[2], comparison results and discussions. Experiment results show our GraphSeq2Seq are better than existing methods on the three benchmarks. The variant model is mostly better than existing methods. Furthermore, we also demonstrate the benefit of combining Graph2Sqe (Gildea et al., 2018) and Seq2Seq (Bahdanau et al.,

---

[2]We will release our code on Github upon the acceptance.

Table 1: Performance comparison on IWSLT2014 German-English dataset.

| Method | BLEU | |
|---|---|---|
| | Greedy Search | Beam search |
| MIXER (Ranzato et al., 2015) | 20.73 | 21.83 |
| BSO (Wiseman & Rush, 2016) | 23.83 | 25.48 |
| LL (Bahdanau et al., 2017) | 26.17 | 27.61 |
| RF-C+LL (Bahdanau et al., 2017) | 27.70 | 28.30 |
| AC+LL (Bahdanau et al., 2017) | 27.49 | 28.53 |
| NPMT (Huang et al., 2018) | 28.57 | 29.92 |
| NPMT+LM (Huang et al., 2018) | N/A | 30.08 |
| Coaching GBN (Chen et al., 2018) | N/A | 30.18 |
| Seq2Seq (Bahdanau et al., 2014) | 26.90 | 28.79 |
| Graph2Seq (Gildea et al., 2018) | 19.89 | 22.31 |
| GraphSeq2Seq (Ours) | **29.06** | **30.66** |
| GraphSeq2Seq-Variant (Ours) | 28.21 | 29.61 |

2014). Besides, our GraphSeq2Seq is trained faster than the state-of-the-art method NMPT (Huang et al., 2018).

## 4.1 IWSLT 2014 GERMAN-TO-ENGLISH

This subsection presents the detailed implementations and evaluation results on the IWSLT 2014 German-to-English machine translation benchmark (Cettolo et al., 2014). Its dataset consists of 153K training samples, 7K validation samples, and 7K test samples. We use the same procedure for preprocessing and dataset splits as in Gildea et al. (2018); Huang et al. (2018). The German and English vocabulary sizes are 32,010 and 22,823 as in Huang et al. (2018). With the words that are out of the vocabularies, we leverage an <UNK> symbol to replace them.

**Implementation** We use Tensorflow[3] to implement the proposed models. All input sentences are padded to the same length with an additional mask variable storing the real length of each input. The hidden vector size for sub-graph encoder is 300. Three layers of the highway network (Srivastava et al., 2015) are set to transform and carry the initial node representation. One layer of the bidirectional LSTM (Schuster & Paliwal, 1997) is used for sequence encoding. For the decoder, two layers of LSTM are used to decode the passed informationThe regularization dropout with probability 0.5 is set as in Huang et al. (2018). We trained our models with Adam (Kingma & Ba, 2014) with an initial learning rate 0.001. The batch size is set to 32, and for decoding, we use greedy search and beam search with a beam size of 10 as in Huang et al. (2018).

**Comparison Results** We compare our GraphSeq2Seq model and its variant with existing methods including MIXER (Ranzato et al., 2015), BSO (Wiseman & Rush, 2016), LL (Bahdanau et al., 2017), RF-C+LL (Bahdanau et al., 2017), AC+LL (Bahdanau et al., 2017), NPMT (Huang et al., 2018), NPMT+LM (Huang et al., 2018), Coaching GBN (Chen et al., 2018), Seq2Seq (Bahdanau et al., 2014), and Graph2Seq (Gildea et al., 2018). We leverage BLEU (Papineni et al., 2002) which is a method for automatic evaluation of machine translation to evaluate our models. The higher the BLEU score is, the better the translations are. Table 1 summaries the performance comparison on IWSLT2014 German-to-English dataset. GraphSeq2Seq achieves better results than the compared methods. Compared with the state-of-the-art methods NPMT and NPMT+LL (Huang et al., 2018), GraphSeq2Seq achieves 0.49 BLEU gains in the greedy setting and 0.58 BLEU gains using beam search. Even GraphSeq2Seq does not get much gains, it is much faster than NPMT and NPMT+LM. It takes about 16 hours to run to convergence (22 epochs) on a machine with one TITAN X GPU, while Huang et al. (2018) notes that NPMT takes about 2-3 days to run to convergence (40 epochs) on four M40 GPUs. Therefore, our GraphSeq2Seq is at least 8 times faster than NPMT. Besides, Huang et al. (2018) also reports the state-of-the-art performance on this dataset, and our GraphSeq2Seq achieves 0.48 BLEU gains on beam search. Comparing with Seq2Seq and Graph2Seq, GraphSeq2Seq achieves at least 2.16 gains on greedy search and 1.87 gains using beam search, which significantly demonstrates the effectiveness of the combination of Seq2Seq and Graph2Seq. For the variant of GraphSeq2Seq, it is slightly lower than GraphSeq2Seq but achieves comparable performance with NPMT.

---

[3]https://www.tensorflow.org

Table 2: Performance comparison on
IWSLT2014 English-to-German dataset.

| Method | BLEU | |
|---|---|---|
| | Greedy | Beam |
| NPMT (Huang et al., 2018) | 23.62 | 25.08 |
| NPMT+LM (Huang et al., 2018) | N/A | 25.36 |
| Seq2Seq (Bahdanau et al., 2014) | 21.26 | 22.59 |
| Graph2Seq (Gildea et al., 2018) | 20.32 | 22.39 |
| GraphSeq2Seq (Ours) | **26.02** | **27.32** |
| GraphSeq2Seq-Variant (Ours) | 25.78 | 27.00 |

Table 3: Performance comparison on
IWSLT2015 English-to-Vietnamese dataset.

| Method | BLEU | |
|---|---|---|
| | Greedy | Beam |
| Hard monotonic (Raffel et al., 2017) | 23.0 | N/A |
| Luong & Manning (2015) | N/A | 23.3 |
| NPMT (Huang et al., 2018) | 26.91 | 27.69 |
| NPMT+LM (Huang et al., 2018) | N/A | 28.07 |
| Seq2Seq (Bahdanau et al., 2014) | 25.50 | 26.10 |
| Graph2Seq (Gildea et al., 2018) | 22.70 | 24.73 |
| GraphSeq2Seq (Ours) | 28.44 | 29.25 |
| GraphSeq2Seq-Variant (Ours) | **28.48** | **29.62** |

## 4.2 IWSLT 2014 ENGLISH-TO-GERMAN

**Implementation** For the IWSLT 2014 English-to-German machine translation benchmark, following the setup presented in Section 4.1, the same dataset is used but with the opposition direction. We utilize the same settings for our model as the German-to-English task, including the batch size, beam search size, optimization algorithm, dropout probability, and hidden layer sizes, etc.

**Comparison Results** We compare with Seq2Seq (Bahdanau et al., 2014), Graph2Seq (Gildea et al., 2018), and state-of-the-art methods NPMT and NPMT+LM (Huang et al., 2018) on this dataset. Table 2 reports their performance comparison. It shows GraphSeq2Seq achieves better results. Compared with the state-of-the-art methods NPMT and NPMT+LL, GraphSeq2Seq achieves 2.4 BLEU gain on greedy search and 1.96 BLEU gain using beam search. The variant model also outperforms NPMT and NPMT+LL by 2.16 BLEU gain and 1.64 BLEU gain on greedy and beam search respectively. GraphSeq2Seq is much better than Seq2Seq and Graph2Seq which demonstrates the effectiveness of our contribution that combines Seq2Seq and Graph2Seq. Besides, our GraphSeq2Seq takes about 15 hours to run to convergence (19 epochs) on a machine with one TITAN X GPU, while Huang et al. (2018) notes that NPMT still takes about 2-3 days to run to convergence (40 epochs) on four M40 GPUs. On this dataset, GraphSeq2Seq is still at least 8 times faster than NPMT.

## 4.3 IWSLT 2015 ENGLISH-TO-VIETNAMESE

**Implementation** For IWSLT 2015 English-to-Vietnamese machine translation benchmark, the dataset consists of roughly 133K training samples, 15K validation samples (from TED tst2012) and 13K test samples (from TED tst2013). The same preprocessing, vocabularies, dropout probability, and batch size are set as in Huang et al. (2018). The other hyperparameters are the same as presented in Section 4.1.

**Comparison Results** We compare our GraphSeq2Seq with Hard monotonic (Raffel et al., 2017), Luong & Manning (2015), Seq2Seq (Bahdanau et al., 2014), Graph2Seq (Gildea et al., 2018), and state-of-the-art methods NPMT and NPMT+LM (Huang et al., 2018). Hard monotonic is an end-to-end differentiable method for learning monotonic alignments which, at test time, enables computing attention online and in linear time. Luong & Manning (2015) explored the use of Neural Machine Translation and demonstrated that an off-the-shelf NMT framework can achieve competitive performance with very little data. Table 3 reports the performance comparison. It shows GraphSeq2Seq performs much better than the compared methods. Compared with the state-of-the-art methods NPMT and NPMT+LL, GraphSeq2Seq achieves 1.53 BLEU gain on greedy search and 1.18 BLEU gain using beam search. The variant model also outperforms NPMT and NPMT+LL by 1.57 BLEU gain and 1.55 BLEU gain on greedy search and beam search respectively. Our GraphSeq2Seq is much better than Seq2Seq and Graph2Seq which verifies the contribution of their combination.

## 4.4 WMT 2016 ENGLISH-TO-CZECH

**Implementation** WMT 2016 English-to-Czech machine translation benchmark, the dataset consists of roughly 181K training samples, 27K validation samples and 30K test samples. The same preprocessing is leveraged but here we use BPE on the target side as Bastings et al. (2017). The vocabulary sizes of the source side and target side are 33,786 and 8,000 respectively. The other hyperparameters are the same as presented in Section 4.1.

Table 4: Performance comparison on WMT 2016 English-Czech dataset.

| Method | BLEU$_1$ | BLEU |
|---|---|---|
| BoW+GCN (Bastings et al., 2017) | 35.4 | 7.5 |
| CNN+GCN (Bastings et al., 2017) | 36.1 | 8.7 |
| BiRNN+GCN (Bastings et al., 2017) | 38.8 | 9.6 |
| Seq2Seq (Bahdanau et al., 2014) | 38.2 | 9.93 |
| Graph2Seq (Gildea et al., 2018) | 36.0 | 8.22 |
| GraphSeq2Seq (Ours) | **40.2** | **11.11** |
| GraphSeq2Seq-Variant (Ours) | 39.6 | 10.7 |

**Comparison Results** Bastings et al. (2017) provide three baselines for Graph Convolutional Networks (GCN), each with a different encoder: a bag-of-words encoder, a convolutional encoder, and a BiRNN. They use predicted syntactic dependency trees of source sentences to produce representations of words that are sensitive to their syntactic neighborhoods. As shown in Table 4, our GraphSeq2Seq performs better than the compared methods. Compared with the state-of-the-art method BiRNN+GCN (Bastings et al., 2017), GraphSeq2Seq increases the BLEU scores by 1.4 (BLEU$_1$) and 1.51 (BLEU), where the BLEU$_1$ means the score for 1-grams and BLEU is for up to 4-grams. Our GraphSeq2Seq is much better than Seq2Seq and Graph2Seq which demonstrates the effectiveness of their combination.

## 4.5 DISCUSSIONS

This subsection is to discuss the effectiveness of our GraphSeq2Seq model including 1) quantitative analysis on graph and sequence information, 2) the impact of highway layers, and 3) the impact of the weight in sub-graph encoder.

**Quantitative analysis on graph and sequence information** This experiment is used to verify the quantitative analysis of our GraphSeq2Seq with random graph and sequence noises based on IWSLT2014 German-English dataset. As shown in Table 5, the random noises change from 0% to 75%, where 75% indicates that 75% of graph and sequence information are noises. 100% is not performed because it is meaningless in real life. Table 5 shows that the BLEU scores go to bad from 29.06 (Greedy) and 30.66 (Beam) to 17.38 and 20.28 when the sequence noise varies from 0% to 75%. For the graph noise, we have a similar observation that the BLEU scores go to bad from 29.06 (Greedy) and 30.66 (Beam) to 24.19 and 26.08 when the graph noise varies from 0% to 75%. It demonstrates that both graph and sequence information are effective in our GraphSeq2Seq, and the performance relies on their qualities.

Table 5: Quantitative analysis of our GraphSeq2Seq on BLEU scores with random graph and sequence noises based on IWSLT2014 German-English dataset. The random noises change from 0% to 75%. Note that 75% indicates that 75% of graph and sequence information are noises.

| | | 0% | 25% | 50% | 75% |
|---|---|---|---|---|---|
| Graph Noises | Greedy | 29.06 | 27.61 | 24.96 | 24.19 |
| | Beam | 30.66 | 29.22 | 26.87 | 26.08 |
| Sequence Noises | Greedy | 29.06 | 26.11 | 24.09 | 17.38 |
| | Beam | 30.66 | 27.97 | 25.58 | 20.28 |

**The impact of highway layers** In GraphSeq2Seq model, we leverage a highway network (Srivastava et al., 2015) to transform and carry the initial information for input words as presented in Section 3.1. It is necessary to discuss whether the highway layers have contributions to the performance. Therefore, we conduct several experiments with different settings of highway layers. Table 6 reports the performance of GraphSeq2Seq on test set and it is training with different numbers of highway layers. Compared with no highway layers, GraphSeq2Seq using one highway layer improves BLEU from 28.13 and 29.78 to 28.84 and 30.13 respectively on greedy and beam search settings. Nearly 0.5 increment on BLEU demonstrates fusing highway layers contributes to the final performance. We test GraphSeq2Seq by varying layer numbers from 0 to 5 and find using 3 highway layers achieves the best performance. With too many highway layers, the performance may be

Table 6: The impact of highway layers on the performance (BLEU) of GraphSeq2Seq on IWSLT2014 German-English dataset. GraphSeq2Seq is training with different numbers of highway layers.

| | Highway Layers | 0 | 1 | 2 | 3 | 4 | 5 |
|---|---|---|---|---|---|---|---|
| BLEU | Greedy Search | 28.13 | 28.84 | 28.72 | 29.06 | 28.43 | 26.76 |
| | Beam Search | 29.78 | 30.13 | 30.00 | 30.66 | 30.02 | 28.03 |

Table 7: The impact of the weight for graph encoding on the performance (BLEU) of GraphSeq2Seq.

| | German-English | | English-German | | English-Vietnamese | |
|---|---|---|---|---|---|---|
| | *w/o* weight | *w/* weight | *w/o* weight | *w/* weight | *w/o* weight | *w/* weight |
| Greedy Search | 27.59 | **28.84** | 25.11 | **26.02** | 26.65 | **28.45** |
| Beam Search | 29.13 | **30.13** | 26.60 | **27.32** | 28.35 | **29.26** |

worse than that without highway layers. A plausible reason is that using too many highway layers increases so many parameters that it needs to carefully adjust the hyperparameters and needs more epochs to run to convergence.

**The impact of the weight in sub-graph encoder** A weight is used in our sub-graph encoder to learn the incoming and outgoing hidden features as presented in Eq. 13 and 14. Here we discuss whether the weight contributes to the performance. Table 7 shows the performance comparisons for GraphSeq2Seq with or without this weight based on the same hyperparameters. Experiment results on three translation datasets shows that performances of using the weight are much better than the performances that without the weight. It almost improves 1 BLEU score which is a relatively big contribution to the performance of GraphSeq2Seq.

## 5 CONCLUSION AND FUTURE WORK

We proposed GraphSeq2Seq, a machine translation model that combines the Seq2Seq model and Graph2Seq model. Besides the effective sequential modeling, it utilizes the dependency graph which contains additional semantic information for sentence modeling. Experiment results show promising performance of our GraphSeq2Seq on IWSLT2014 English-to-German, German-to-English, IWSLT2015 English-to-Vietnamese, and WMT 2016 English-to-Czech machine translation benchmarks. In performance comparison, GraphSeq2Seq is much better than Seq2Seq and Graph2Seq demonstrating the effectiveness of their combination. It also outperforms the state-of-the-art methods NPMT (Huang et al., 2018), BiRNN+GCN (Bastings et al., 2017) and Coaching GBN (Chen et al., 2018) as well as being trained almost 8 times faster than NPMT.

In future work, we will 1) apply GraphSeq2Seq to larger datasets and more natural language generation tasks; 2) explore the graph structure among words and sentences to address the paragraph-level translation task.

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
