# OpenReview forum: "GraphSeq2Seq: Graph-Sequence-to-Sequence for Neural Machine Translation"
_ICLR.cc/2019/Conference_

### Official Review · AnonReviewer2 · 2018-11-02
**Review of "GraphSeq2Seq: Graph-Sequence-to-Sequence for Neural Machine Translation"**

**Rating:** 6
**Confidence:** 3

**Review:**

This paper proposes a method for combining the Graph2Seq and Seq2Seq models into a unified model that captures the benefits of both.  The paper thoroughly describes in series of experiments that demonstrate that the authors' proposed method outperforms several of the other NMT methods on a few translation tasks.

I like the synthesis of methods that the authors' present.  It is a logical and practical implementation that seems to provide solid benefits over the existing state of the art.  I think that many NMT researchers will find this work interesting.

Table 4 begs the question, "How does one choose the number of highway layers?"  I presume that the results in that table are from the test data set.  Using the hold out data set, which number gives the best value?

The paper's readability suffers from poor grammar in some places.  This fact may discourage some readers.

The authors should fix the missing parentheses in Eqns. (6)-(9).

---

> ### Author Response · Authors · 2018-11-26
> **Response to AnonReviewer2**
>
> Thank you for your friendly and valuable comments.
>
> Q1. The number of highway layers.
> For Table 4, it shows the performance on the test set and it is training with different numbers of highway layers, i.e., it uses the hold out data set. Thus, we conclude that using 3 highway layers achieves the best performance on IWSLT-2014 German-English dataset.
>
> Q2. Poor grammar and missing parentheses.
> We carefully proofread our paper and try to improve the grammar for a good readability. We also fix the missing parentheses in Eqns. (6)-(9).

---

### Official Review · AnonReviewer3 · 2018-11-02
**Promising results, but use of dependency parser is somewhat concerning**

**Rating:** 6
**Confidence:** 4

**Review:**

This paper proposes a seq2seq model which incorporates dependency parse information from the source side by embedding each word's subgraph (according to a predetermined dependency parse) using the Graph2Seq model of Song et al. (2018); the authors propose two variants for achieving an encoded source-side representation from the subgraph embeddings, involving bidirectional lstms. The authors show that the proposed approach leads to good results on three translation datasets.

The paper is generally written fairly clearly, though I think the clarity of section 3.3 could be improved; it took me several reads to understand the architectural difference between this second variant and the original one. The results presented are also impressive: I don't think the IWSLT de-en results are in fact state of the art (e.g., Edunov et al. (NAACL 2018) and Deng et al. (NIPS 2018) outperform these numbers, though both papers use BPE, whereas I assume the current paper does not), but the results on the other two datasets appear to be.

Regarding the approach in general, it would be nice to see how much it depends on the quality of the dependency parse. In particular, while we might expect the en-de and en-vi results to be good because dependency parsers for English are relatively good, how much does performance degrade when considering languages with less good dependency parsers?

Pros:
- Good results, fairly simple model

Cons:
- Somewhat incremental, not clear how much method depends on quality of the dependency parser

---

> ### Author Response · Authors · 2018-11-27
> **Response to AnonReviewer3**
>
> Thank you for your valuable comments and suggestions.
>
> Q1. The clarity of section 3.3.
> In section 3.3, we mainly present a variant of the encoder part of our GraphSeq2Seq. As shown in Fig.2, after the sub-graph encoder, we get the hidden feature for each node. Then we rebuild the sub-graph for the current node. For the rebuilt sub-graph, its outgoing hidden feature is the input of a Graph_Out Bi-LSTM, while its incoming hidden feature is for a Graph_in Bi-LSTM. For the current node, its node representation is utilized for a Node Sequence Bi-LSTM.
> The main difference is that the original model is to encode the concated representation of the sub-graph by using only one Bi-LSTM, while the variant model leverages three Bi-LSTMs to respectively encode the specific representations of the sub-graph, including the incoming feature, the outgoing feature, and the node representation.
> We clarify it in the revision and revise Fig. 2 to illustrate it.
>
> Q2. State-of-the-arts with BPE.
> Thanks for your suggestion. We compare our model with NMPT and NMPT+LM, which are proposed by ICLR18 paper [ref-4]. Both our model and the compared method utilize the same settings and do not use BPE, so they are comparable. We think it can be acceptable. We also would like to compare with Edunov et al. [ref-1] and Deng et al. [ref-2]. However, they utilized BPE on both the source side and target side, whereas our GraphSeq2Seq cannot apply BPE on the source side. Thus, we do not choose this kind of methods for comparison. We will cite and discuss these works [ref-1,ref-2] in the revision.
> Fortunately, we add a comparison with Bastings et al. [ref-3], which utilizes BPE on only the target side. We get 40.2 (BLEU1) and 11.11 (BLEU4) on WMT16 English-Czech dataset, which improves the BLEU scores by 1.4 (BLEU1) and 1.51 (BLEU4) than Bastings et al. [ref-3]. This improvement verifies the effective performance of the proposed method on the top-line NMT baselines. The experiment details are shown in Section 4.4.
>
> Q3. How much it depends on the quality of the dependency parse.
> With regard to the impact of the dependency parse on performance, we add an experiment to discuss it. We randomly add some noise to the parsing result, and then train our model. We find the BLEU scores go to bad when the parsing result contains more noises. That means the BLEU scores degrade when considering languages with less good dependency parsers. More experiments are shown in the Response of Q3 for AC comments and Table 5 in the revision.
>
> [ref-1] Sergey Edunov, Myle Ott, Michael Auli, David Grangier, Marc'Aurelio Ranzato. Classical Structured Prediction Losses for Sequence to Sequence Learning. NAACL-HLT 2018: 355-364.
> [ref-2] Yuntian Deng, Yoon Kim, Justin Chiu, Demi Guo, Alexander M. Rush: Latent Alignment and Variational Attention. NIPS 2018.
> [ref-3] Joost Bastings, Ivan Titov, Wilker Aziz, Diego Marcheggiani, and Khalil Sima’an. Graph convolutional encoders for syntax-aware neural machine translation. In Proc. EMNLP, pp. 1957–1967, 2017.
> [ref-4] Po-Sen Huang, Chong Wang, Sitao Huang, Dengyong Zhou, and Li Deng. Towards neural phrase-based machine translation. In International Conference on Learning Representations, 2018.

---

### Official Review · AnonReviewer1 · 2018-11-03
**Interesting idea**

**Rating:** 6
**Confidence:** 5

**Review:**



[Summary]
This paper proposes a Graph-Sequence-to-Sequence (GraphSeq2Seq) model to fuse the dependency graph among words into the traditional Seq2Seq framework.



[clarity]
This paper is basically well written though there are several grammatical errors (I guess the authors can fix them).
Motivation and goal are clear.


[originality]
Several previous methods have already tackled to integrate graph structures into seq2seq models.
Therefore, from this perspective, this study is incremental rather than innovative.
However, the core idea of the proposed method, that is, combining the word representation, sub-graph state, incoming and outgoing representations seems to be novel.



[significance]
The experimental setting used in this paper is slightly out of the current main stream of NMT research.
For example, the current top-line NMT systems uses subword unit for input and output sentences, but this paper doesn’t.
Moreover, the experiments were performed only on the very small datasets, IWSLT-2014 and 2015, which have at most 153K training parallel sentences.
Therefore, it is unclear whether the proposed method has essential effectiveness to improve the performance on the top-line NMT baselines.

Comparing on the small datasets, the proposed method seems to significantly improve the performance over current best results of NPMT+LM.



Overall, I like the idea of utilizing sub-graphs for simplicity and saving the computational cost to encode a structural (grammatical or semantic) information.
However, I really wonder if this type of technique really works well on the large training datasets...

---

> ### Author Response · Authors · 2018-11-26
> **Response to AnonReviewer1**
>
> Thank you for your kind summarization and valuable comments.
>
> Q1. Using BPE in our experiments.
> Thanks for your suggestion again. We admit that the setting is slightly out of the current main stream of NMT research. To enrich our experiments, we accept your suggestion by adopting BPE to compare with the paper of Bastings et al. [ref-1] which also leverages BPE. We get 40.2 (BLEU1) and 11.11 (BLEU4) on WMT16 English-Czech dataset, which improves the BLEU scores by 1.4 (BLEU1) and 1.51 (BLEU4) than Bastings et al. [ref-1]. This improvement verifies the effective performance of the proposed method on the top-line NMT baselines. The experiment details are shown in Section 4.4.
>
> Q2. Large training datasets.
> We would like to evaluate our model on the large WMT dataset (the WMT2014 English-to-German) but it is still running. After getting the results, we will post the results here and release the full experiments in the final version. For the dataset, the compared methods including ICLR18 [ref-2], NIPS18 [ref-3], EMNLP16 [ref-4], ICML17 [ref-5] only utilize the IWSLT-2014 and 2015 datasets for NMT task. To academic research, we think our experiments on current four datasets from IWSLT-2014 German-to-English, IWSLT-2014 English-to-German, IWSLT-2015 English-to-Vietnamese, and WMT-2016 English-Czech can be acceptable.
>
> [ref-1] Joost Bastings, Ivan Titov, Wilker Aziz, Diego Marcheggiani, and Khalil Sima’an. Graph convolutional encoders for syntax-aware neural machine translation. In Proc. EMNLP, pp. 1957–1967, 2017.
> [ref-2] Po-Sen Huang, Chong Wang, Sitao Huang, Dengyong Zhou, and Li Deng. Towards neural phrase-based machine translation. In International Conference on Learning Representations, 2018.
> [ref-3] Minjia Zhang, Xiaodong Liu, Wenhan Wang, Jianfeng Gao, Yuxiong He. Navigating with Graph Representations for Fast and Scalable Decoding of Neural Language Models, NIPS, 2018.
> [ref-4] Sam Wiseman and Alexander M. Rush. Sequence-to-sequence learning as beam-search optimization. In Proc. EMNLP, pp. 1296–1306, 2016.
> [ref-5] Colin Raffel, Minh-Thang Luong, Peter J. Liu, Ron J. Weiss, and Douglas Eck. Online and linear- time attention by enforcing monotonic alignments. In Proc. ICML, pp. 2837–2846, 2017.

---

### Comment · Area_Chair1 · 2018-11-09
**Clarification about comparison to baselines?**

This paper proposes a new method for incorporating graph structures in sequence-to-sequence models. The idea itself seems reasonable, but the comparison to the baseline is concerning.

First, just looking at the normal comparison on IWSLT2014, the seq2seq and graph2seq baselines only achieve BLEU scores of 23.87 and 22.31 respectively, which is not at all competitive with the state-of-the-art. Just well-tuned sequence-to-sequence models can achieve a score of 29.10 on this dataset (see "Generative Bridging Network for Neural Sequence Prediction", NAACL 2018). I fear that the baselines here are too weak, and the comparison with them too indirect to really tell us anything about the merit of the proposed model.

Second, the graph2seq model used as a baseline was not designed as a method for MT, but rather for logical-form-to-text generation. A method such as that of Bastings et al., which was specifically designed for MT, seems to be a more fair comparison.

Third, there is no quantitative analysis or qualitative comparison of why or how the proposed method outperforms the graph2seq baselines. It would be nice to know more about why the proposed methods are helping compared to other reasonable methods.

I'd appreciate if the authors could clarify about these concerns in their response.

---

> ### Author Response · Authors · 2018-11-27
> **Response to Area Chair1**
>
> Thank you for your valuable comments and suggestions.
>
> Q1. The weak baseline.
> For the Seq2Seq baseline, we report the BLEU scores of 23.87 and 22.31, which are from the HarvardNLP work [ref-1]. Because they focus on the beam-search optimization, the non-fine-tuning results in the low BLEU scores. After fine-tuning Seq2Seq, we get the BLEU scores of 28.79 and 26.90 on beam and greedy search respectively. We also compare with the NAACL18 work [ref-2], which is still lower than our GraphSeq2Seq by 0.48. Compared to Seq2Seq, we mainly show that Graph2Seq can be used to improve the performance of Seq2Seq. This also reveals the effectiveness of our GraphSeq2Seq since it is a reasonable combination of Graph2Seq and Seq2Seq. All results are updated in the revision.
>
> Q2. Fairness for Graph2Seq.
> To fairly compare with Bastings et al. [ref-3] which also fuses graph information for the NMT task, we set the same settings to do experiments. Our GraphSeq2Seq gets 40.2 (BLEU1) and 11.11 (BLEU4) on WMT16 English-Czech dataset, whereas the results of [ref-3] are 38.8 (BLEU1) and 9.6 (BLEU4). That means GraphSeq2Seq increases the BLEU scores by 1.4 (BLEU1) and 1.51 (BLEU4) than Bastings et al. [ref-3]. The experiment details are shown in Section 4.4.
> Moreover, NIPS’18 [ref-4] also considers Graph Representations in NMT task. By revising our vocab size to 50,000 trying to fairly compare with [ref-4], we get the BLEU scores of 29.47 and 27.90 on beam and greedy search on IWSLT2014 German-English dataset, which are slightly higher by 0.13 and 0.27 than the result reported by [ref-4].
>
> Q3. Quantitative analysis of how GraphSeq2Seq outperforms Graph2Seq and Seq2Seq.
> We do two experiments to show the quantitative analysis of our GraphSeq2Seq.
> 1) The first experiment is used to show the impact of the sequential encoder in the Graph2Seq [ref-5] based on three datasets. To a fair qualitative comparison, we directly use the code of the Graph2Seq [ref-5]. By adding the bidirectional sequence encoder into this code, our GraphSeq2Seq gets the improvements of 8.35 and 9.17 on beam and greedy search on IWSLT2014 German-English dataset,  4.93 and 5.7 on IWSLT2014 English-German dataset, 4.89 and 5.78 on IWSLT2015 English-to-Vietnamese dataset, 4.2 (BLEU1) and 2.89(BLEU4) on WMT2016 English-to-Czech dataset. This significant improvement definitely verifies the effectiveness of the sequential encoder. Furthermore, based on Seq2Seq, adding the sub-graph encoder gets the improvements of 1.87 and 2.16 on beam and greedy search on IWSLT2014 German-English dataset,  4.73 and 4.76 on IWSLT2014 English-German dataset, 3.52 and 2.98 on IWSLT2015 English-to-Vietnamese dataset, 2.0 (BLEU1) and 1.18(BLEU4) on WMT2016 English-to-Czech dataset. This significant improvement definitely verifies the effectiveness of the sub-graph encoder. The results of this experiment are reported in Tables 1, 2, 3, and 4.
> 2) The second experiment is used to verify the quantitative analysis of our GraphSeq2Seq with random graph and sequence noises based on IWSLT2014 German-English dataset. As shown in Table 2, the random noises change from 0% to 75%, where 75% indicates that 75% of the graph and sequence information are noises. 100% is not performed because it is meaningless in real life. Table 2 shows that the BLEU scores go to bad from 29.06 (Greedy) and 30.66 (Beam) to 17.38 and 20.28 when the sequence noise varies from  0% to 75%. For the graph noise, we have a similar observation that the BLEU scores go to bad from 29.06 (Greedy) and 30.66 (Beam) to 24.19 and 26.08 when the graph noise varies from  0% to 75%. It demonstrates that both graph and sequence information are effective in our GraphSeq2Seq, and the performance relies on their qualities. The detailed experiment is shown in Table 5 in the revision.
>
> [ref-1] Sam Wiseman and Alexander M. Rush. Sequence-to-sequence learning as beam-search optimization. In Proc. EMNLP, pp. 1296–1306, 2016.
> [ref-2] Wenhu Chen, Guanlin Li, Shuo Ren, Shujie Liu, Zhirui Zhang, Mu Li, Ming Zhou: Generative Bridging Network for Neural Sequence Prediction. NAACL-HLT 2018: 1706-1715.
> [ref-3] Joost Bastings, Ivan Titov, Wilker Aziz, Diego Marcheggiani, and Khalil Sima’an. Graph convolutional encoders for syntax-aware neural machine translation. In Proc. EMNLP, pp. 1957–1967, 2017.
> [ref-4] Minjia Zhang, Xiaodong Liu, Wenhan Wang, Jianfeng Gao, Yuxiong He. Navigating with Graph Representations for Fast and Scalable Decoding of Neural Language Models, NIPS, 2018.
> [ref-5] Daniel Gildea, Zhiguo Wang, Yue Zhang, and Linfeng Song. A graph-to-sequence model for amr- to-text generation. In Proc. ACL, pp. 1616–1626, 2018.

---

### Meta-Review · Area_Chair1 · 2018-12-13
**Interesting method, if somewhat incremental. Experiments are reasonable but variables potentially not controlled.**

**Confidence:** 2
**Recommendation:** Reject

**Metareview:**

This paper proposes a new method for graph representation in sequence-to-sequence models and validates its results on several tasks. The overall results are relatively strong.

Overall, the reviewers thought this was a reasonable contribution if somewhat incremental. In addition, while the experimental comparison has greatly improved from the original version, there are still a couple of less satisfying points: notably the size of the training data is somewhat small. In addition, as far as I can tell all comparisons with other graph-based baselines actually aren't implemented in the same toolkit with the same hyperparameters, so it's a bit difficult to tell whether the gains are coming from the proposed method itself or from other auxiliary differences.

I think this paper is very reasonable, and definitely on the borderline for acceptance, but given the limited number of slots available at ICLR this year I am leaning in favor of the other very good papers in my area.